# Reduction in Blood Lead Concentration in Children across the Republic of Georgia following Interventions to Address Widespread Exceedance of Reference Value in 2019

**DOI:** 10.3390/ijerph182211903

**Published:** 2021-11-12

**Authors:** Ekaterine Ruadze, Giovanni S. Leonardi, Ayoub Saei, Irma Khonelidze, Lela Sturua, Vladimer Getia, Helen Crabbe, Tim Marczylo, Paolo Lauriola, Amiran Gamkrelidze

**Affiliations:** 1The National Center for Disease Control and Public Health of Georgia, Tbilisi 0198, Georgia; i.khonelidze@ncdc.ge (I.K.); lela.sturua@ncdc.ge (L.S.); Kh.getia@ncdc.ge (V.G.); a.gamkrelidze@ncdc.ge (A.G.); 2UK Health Security Agency, Radiation, Chemical and Environmental Hazards, Harwell Campus, Didcot OX11 0RQ, UK; Giovanni.leonardi@phe.gov.uk (G.S.L.); helen.crabbe@phe.gov.uk (H.C.); tim.marczylo@phe.gov.uk (T.M.); 3Department of Social and Environmental Research, London School of Hygiene and Tropical Medicine, London WC1E 7HT, UK; 4UK Health Security Agency, Statistics Unit, Department of Statistics, Modelling and Economics, London NW9 5EQ, UK; Ayoub.Saei@phe.gov.uk; 5International Society of Doctors for the Environment-Italy (ISDE-Italy), 42122 Modena, Italy; paolo.lauriola@gmail.com

**Keywords:** lead (Pb), Georgia, public health interventions, state program, written and verbal communication, multiple stakeholder response

## Abstract

In recent years, reports of lead contamination have dramatically increased in Georgia. Given concerns about the exposure of children to lead (Pb), the National Multiple Indicator Cluster Survey (MICS-2018) included a blood sampling component. The results showed that 41% of the children that participated had blood Pb levels (BLL) ≥ 5 µg/dL and that BLL in children living in Western Georgia were higher than those in Eastern regions. In response to these findings, NCDC implemented written and verbal advice to the families of children who participated in the MICS-2018 on how to reduce Pb exposure. From August 2019 onwards, the state program of clinical follow-up was implemented. The design of this study was a longitudinal study. The intervention of interest was the public health advice and medical follow-up, and the outcome was defined as the difference in BLL between the MICS-2018 survey and the state program follow-up. We observed a significant overall reduction in median BLL between MICS-2018 and state program follow-up in both August 2019 and the latest results (until December 2019). However, we did not observe any significant further reduction between August and the most recent BLL results. In the Georgian setting, written and verbal communication targeting individual households, alongside home visits to the most exposed, effectively reduced BLL in children.

## 1. Introduction

Lead (Pb) is a widespread and harmful environmental toxicant [1,2,3] that causes adverse health effects in children, particularly neurological and neurobehavioral deficits, lower IQ, slowed growth and anemia [4,5,6,7]. In addition, the health effects of Pb over the life course have been documented, including adverse effects on the cardiovascular [8,9], renal and hepatic systems [10,11,12].

In recent years, reports of lead poisoning/contamination have dramatically increased in Georgia [13]. In response, Georgia’s National Center for Disease Control and Public Health (NCDC) and the U.S. Centers for Disease Control (CDC) office in Tbilisi conducted a small-scale survey in November−December 2015. Of the two hundred and fifty-four (254) children, aged 2–5 years, tested at The Iashvili Children’s Hospital in Tbilisi, around 46% of children had a blood lead level (BLL) exceeding the national action level (5 µg/dL). Thirty-three percent (33%) of survey participants had BLL ≥ 5 µg/dL, 9.5% ≥ 10 µg/dL, 2.8% ≥ 20 µg/dL and 0.4% ≥ 45 µg/dL [14]. A follow-up study in December 2015 investigated 46 children aged 4–6 years from 10 villages of Bolnisi and Dmanisi districts. This investigation found that 30.4% of participants had BLL ≥ 5 µg/dL.

To better understand the scale of lead (Pb) exposure in Georgian children, UNICEF funded the National Multiple Indicator Cluster Survey (MICS-2018) in 2018, conducted in collaboration with the NCDC, National Statistics Office of Georgia, and Italian Istituto Superiore di Sanitá (ISS). This nationally representative sample of 1578 children aged between 2 and 7 years included determination of BLL. In agreement with the previous studies, the results showed that BLL was elevated in a number of children, with 41% of these children having BLL ≥ 5 µg/dL, with regional range from a minimum of 18% in Kvemo-Kartli in the east of Georgia to a maximum of 85% in Adjara in the west [15].

### Lead in the Environment of Georgia

In 2011, 2015 and 2017, the New York Health Department found significantly high levels of Pb in Georgian spices and in the blood of Georgian expats living in New York [16].

Over a similar period, Tbilisi State University conducted a study in 2017 in Kvemo Kartli districts (Bolnisi and Dmanisi) that showed high contamination of soil by Pb, mercury and cadmium [17]. This pollution has been associated with industrial activity and pollution of water [18].

Routine monitoring data for the period of 2017–2018 of Pb concentrations in soils, obtained from the National Environmental Agency (NEA), showed that out of the 485 soil samples analyzed 22% (105/485) had Pb concentration above the acceptable level (≥32 mg/kg) according to Georgian regulations. Around the same time, routine monitoring data (2017–2019) obtained from the National Food Agency (NFA) showed that Pb levels above the limit values were mainly found in spices.

Furthermore, according to a report by the Center for Strategic Research and Development of Georgia, toys sold in Georgia also contain high levels of Pb and other harmful metals [19], while the International Pollutants Elimination Network (IPEN) also reported high Pb content in the paints sold/used in the country [20].

In light of these findings, a study was conducted (January–March, 2019) with 17 families living in Tbilisi, for whom blood analysis for Pb had been completed, to investigate possible sources of lead exposure. Out of the 268 environmental samples investigated, Pb was found in building materials, toys, spices, cosmetics and other materials [21].

Following the MICS-2018 survey, a range of environmental media was assessed in twenty-five homes and four bazaars, spanning five regions across Georgia. Exceptionally high lead concentrations were identified in multiple spices. Median lead concentrations in six spices with elevated Pb ranged from 4–2418 times higher than acceptable levels. Median lead concentrations of all other media were within internationally accepted guidelines [22].

All accumulated evidence moved public health agencies to establish an initial public health action plan to assess environmental samples and develop a state program for further monitoring of BLL in children.

This study is a preliminary investigation in preparation for others, such as Pb isotope ratio analysis. Ongoing studies are motivated to provide evidence to support the implementation of a universally accepted, specific exposure prevention strategy.

The overall work program has been arranged and carried out in collaboration with international organizations, such as the UK Health Security Agency (UKHSA) (previously Public Health England, PHE) and ISS, to better enable the health authorities in Georgia to address this emergency both in terms of scientific appropriateness and in terms of positive political collaboration.

In particular, the analyses reported in this paper aim to assess the efficacy of the interventions put in place following MICS-2018 by measuring BLL in children previously identified as having elevated BLL in their MICS-2018 sample and provide an initial evaluation of such interventions. Specific objectives were to (i) compare the initial MICS-2018 BLL and the subsequent BLL results from the state program for early disease detection and screening (hereafter “the state program”) and (ii) examine the longitudinal trend in BLL change.

## 2. Materials and Methods

The design was a longitudinal study with an observational design to monitor the changes in BLL following the public health interventions implemented after the MICS-2018 survey identified a large proportion of BLL exceeding the action level, and it was considered as a two-stage intervention in all children included in the MICS-2018 survey and their household.

The first-stage intervention was based on written and verbal communication on reducing Pb exposure by NCDC and its regional partners to the families of children who participated in the MICS-2018, and the second-stage intervention was the state program of clinical follow-up, which was implemented from August 2019 onwards. In the framework of the state program, BLL was tested at least once, and tests were repeated if the child’s BLL was ≥5 μg/dL. The MICS-2018 BLL test results were treated as a baseline.

Therefore, the intervention of interest in these analyses was public health advice. The outcome of this analysis was defined as the difference in BLL between the MICS-2018 survey and later interventions.

### 2.1. NCDC Communication (First-Stage Intervention)

Letters reporting the BLL results and recommendations relevant to reducing Pb exposure at the household level were sent to each of the 1578 families. NCDC, CDC Tbilisi office and the University of Emory were involved in developing these recommendations. The letters outlined the possible sources and health consequences of Pb exposure and provided advice to parents on how to reduce Pb exposure within the household and in particular on dietary habits (increasing intakes of food rich in calcium, iron and vitamin C) that can help to reduce BLL (Table 1).

If the BLL was ≥5 μg/dL, parents were advised to take the child to a pediatrician to assess physical and mental development and iron deficiency, and if required, to follow the pediatrician’s recommendations for further tests. All expenses were covered by the state program described below.

Parents were also told by letter that municipality public health specialists would further investigate their living environment for potential sources of Pb exposure.

### 2.2. The State Program (Second-Stage Intervention)

The state program in 2019 included a BLL biomonitoring component for children who had previously participated in MICS-2018 and also included their family members < 18 years old and pregnant family members [23].

Pediatrician’s consultation within the state program included: (i) assessment of physical and mental development of children with a pre-designed questionnaire; (ii) assessment of nutritional status: vitamins, calcium and iron intake; (iii) provision of information about possible sources of Pb exposures and international recommendations. If BLL was ≥5 µg/dL, the child was provided with iron and calcium supplements and multivitamins [23].

The state program also included the training of family doctors, pediatricians and public health specialists in early detection and management of lead exposure [23]. Each child’s age and place of residence were collected both as part of the MICS-2018 survey and state program [23].

Venous BLLs were measured using a LeadCare II blood lead analyzer provided by Magellan Inc. in the MICS-2018 survey [14] and with an atomic absorption spectrometer in the state program [23].

All children tested in MICS-2018 with a BLL > 5 µg/dL were followed up by the Iashvili children’s hospital clinic team with repeated BLL tests in 2019 which were included in the statistical analysis.

Testing frequencies were dependent upon MICS-2018 BLL. Children whose BLL dropped below the action level were not followed up after this. Children with >10 ug/dL were tested up to 4 times in the state program [23].

### 2.3. Statistical Methods

All children (i) less than 18 years old (ii) tested in the framework of the 2019 state program and (iii) also in the MICS-2018 were included in the statistical analysis. In the framework of the state program were also children (<18 years old) that had not been tested in the MICS-2018 which were excluded from the analysis, as well as adult pregnant family members. For a general description of the cohort, we calculated frequencies.

BLL were measured within three different programs on each child in the study. Children were clustered within 10 different geographical regions, and each region was further grouped into either the West region or the East region. Children residing in the Western regions including Adjara, Guria, Imereti and Samegrelo had statistically higher BLL (Table 2). The repeated measurement and cluster are two essential characteristics of the BLL collected data in this study. Measurements from different children are considered independent, while those from the same child/household are potentially correlated.

A statistical problem facing researchers involved in such studies concerns the proper accounting in analysis of the correlation among measurements taken from the same child. We advanced here a statistical modeling approach that allows the measurements from the same child to be correlated. However, preliminary results (not reported here) were not satisfactory from the direct application of the model to BLL data. After studying the residuals under an initial model, a log transformation was applied to the outcome data (BLL). The log-transformed data were then used as response in modeling. The subdivision of the regions into west and east was used as strata in the model. The model included the region as a random effect in addition to the program. The model allowed the observations to be correlated with the same child. A component symmetry or exchangeable correlation was selected to fit the study design. The MICS-2018 was set up as the baseline/reference category in the model. This immediately implies that the August 2019 and most recent BLL data could be directly compared from model results. Let Y_ijkl denote the log-transformed random variable (BLL) of the lth measurement (l = 1, 2, 3 program), from the kth child (k = 1,…) within the region j (j = 1,…) from the strata I (i = 1 = West, 2 = East). The linear mixed model underlying the data analysis is then given by
Model 1: Y_ijkl = μ + α_i + β_il + u_(j(i)) + e_ijkl(1)
where μ (West) is the intercept, α_2 = 0 and α_1 is the strata effect (East). The β_ik (i = 1, 2 and l = 1, 2 and 3) is the program effect with β_i1 = 0, i.e., the MIC-2018 was set to baseline. The region with random effects, u_(j(i)), is independent normal variable with zero means and common variance ∅. The last term e_ijkl corresponds to the normally distributed error terms. The error terms are allowed to be correlated within children and independent between children. An exchangeable or constant correlation structure was adopted here. The model parameters were estimated using the restricted maximum likelihood (REML) method. The computation was done in SAS using [SAS/STAT] software, Version 9.4 of the SAS 64 BIT WIN [14].

### 2.4. Ethics

According to the Law of Georgia on Patient Rights, written informed consent is required within medical establishments and was obtained here. The current study analysis BLL data were collected in two stages: first in the framework of the MIC-2018 survey, and second in the state program. Written informed consent from children’s parents or guardians was collected for each blood sample, both those included in the MICS-2018 and those covered by the state program. In the framework of the state program, blood was collected in the medical establishments. Parents/guardians were free to withdraw their consent at any time. Ethics approval for MICS-2018 was obtained on the 8 August 2018 (NCDC IRB # 2018-033). The institutional review board approved the presented study in June 2020 (NCDC IRB #2020-026) [24].

## 3. Results

The data set included 423 children, out of which 38% (160 children) were tested for BLL only once, 41% (175 children) were tested twice, 19% (80 children) were tested three times and 1.9% (8 children) were tested four times. The regional distribution of the cohort across the 10 regions of Georgia varied from a minimum of 9 children in Kvemo Kartli to a maximum of 99 children in Guria.

If a child was tested only once (160 children), his/her BLL result was considered to be the most recent result. Out of a total cohort (423 children), 256 were tested in August and after August 2019. For these children, the last BLL results obtained after August (September–December, 2019) was considered to be the most recent result.

We observed the overall reduction of median BLL between MICS-2018 (9.6 µg/dL) and August 2019 (7.1 µg/dL) test results and between MICS-2018 and the most recent BLL results (6.8 µg/dL). However, there was not enough information to support significant changes between the August 2019 and the most recent BLL results. Median BLL with inter-quartile range (IQR) 25–75% is given in Table 2.

The reduction of BLL was statistically significant (*p* < 0.001) both in the Eastern and Western regions of Georgia. The mean reduction is greater between MICS-2018 and the most recent BLL results in both regions compared to MICS-2018 and August 2019 BLL results (Table 3).

However, pairwise comparison (Table 4) did not support statistically significant differences between August 2019 and the most recent BLL results (in either or Eastern Georgia).

There was strong support for a statistically significant lower level of BLL in Eastern compared to Western Georgia (*p* < 0.001).

## 4. Discussion

### 4.1. Key Findings

In the children studied, we observed a median reduction of BLL between MICS-2018 and first-stage intervention in August 2019 by 2.5 µg/dL and between MICS-2018 and the most recent BLL results by 2.8 µg/dL. The median BLL reduction between August 2019 and the most recent BLL results (0.3 µg/dL) was not statistically significant.

### 4.2. Limitations of the Study

We did not evaluate which interventions among those mentioned in the communication letter after MICS-2018 were implemented by the families or whether households chose to implement other interventions not mentioned in the letter. In any case, some key actions recommended by WHO were encouraged in the targeted children’s households. As it was not considered ethical to monitor children with elevated BLL and not provide advice on reducing Pb exposure, we also cannot discount entirely that the decline in BLL is not a consequence of other factors.

The state program interventions in 2019 were targeted at children whose BLL in the MICS-2018 survey was above 5 μg/dL. For this reason, we cannot talk about changes in BLL among children whose test results were below 5 μg/dL. If a child’s BLL, included in the state program 2019, reduced below 5 μg/dL, this child was no longer tested in the framework of the state program. We cannot be certain that BLL in these children would have remained below 5 μg/dL or decreased further, but if it had and the children were included in the latter sampling round, the mean BLL decrease observed at the end of the study period could have been more significant. As 37.8% of children were tested only once in the state program, the sample size for sustainability analysis was reduced and consequently underpowered to detect a statistical difference between August 2019 and last samples. In addition, the sampling period of the state program (August to December 2019) was relatively short to detect further changes of BLL in these children.

### 4.3. Strengths of the Study

Results are based on nationally representative sampling of MICS-2018 survey and returning to the same sampled children for the state program. Another strength of the study is that we used not a subset of the state program participant but the whole dataset.

### 4.4. The Value of the Intervention Provided

Compared with other Pb reduction interventions across a nationally distributed population of children, the reduction in BLL within the present study is larger than that achieved in Canada [25] and similar to that achieved in North Carolina and Vermont [26] in the late 1990s, although the interventions need to be sustained over a period of several years in order to achieve a persistent and more marked reduction in BLL [27].

In our study, we observed a median decline in BLL of 2.5 ug/dL over 8–12 months. This compares favorably with NHANES data [27], where >70% reduction in BLL was observed between the 1976–80 and the 1988–91 surveys, where declines of 15.0 to 3.6 μg/dL and 11.7–1.9 μg/dL for age groups 1–5 y and 6–19 y, respectively, were observed over a minimum of 8 years. A later investigation in North Carolina and Vermont [26] using samples collected between 1996–1999 reported that it required just over 1 year (382 days) for BLL to decline to <10 μg/dL, with the highest BLLs taking even longer in children younger than 5 y reporting BLL > 10 μg /dL. We also report median BLL below this cut-off in all regions in the August 2019 sample, though one region (Adjara) did report a median BLL of 10.2 μg/dL in the most recent blood sample (Table 2).

The reduction in BLL is a substantial improvement within the target population over the period considered. However, achieving a similar magnitude of improvement across the whole population would require a different type of intervention, as a pediatrician-led intervention may not be the most cost-effective. The interventions implemented in Georgia were deployed as part of a program to address a national emergency of widespread children’s BLL greater than reference limit value. This meant that immediate priority was given to the deployment of well-known interventions to reduce Pb exposure, rather than establishing a community randomized control trial (RCT). Reviews of RCTs evaluating interventions to reduce BLL as an indicator of Pb exposure have not previously supported educational interventions alone [28], but education accompanied by provision of assistance with household cleaning may be effective [29], and the need to focus efforts on home visits and Pb source investigations was the conclusion of a review of Pb reduction interventions in Australia [30].

In any case, the reduction in BLL shows that a Pb intervention program can improve BLL in a relatively short period. Potentially, a population intervention based on health promotion aimed at increasing awareness of Pb exposure and harm, as well as behavioral change, could achieve a sustained improvement across the population. In our case, the population-based intervention was targeted and tailored for children who were already tested and identified with elevated BLL concentrations. The approach Georgia used was direct written and verbal communication to each family, which we believe contributed to the significant efficacy of the intervention. A wide range of interventions in and around the house remain recommended by the World Health Organization to address Pb exposure [31].

### 4.5. The Need to Continue Intervention to Reduce Pb Exposure in the Republic of Georgia

The documented high content of Pb in spices commonly used in Georgian cuisine confirms the presence of Pb at concentrations of potential concern in food items as one of the sources of exposure in this population [22]. Although further interventions are expected to have a cost, in general, there are substantial returns to investing in Pb exposure control, mainly targeted at early intervention in communities most likely at risk [32] and prioritizing environmental control measures over clinical management [33]. Although partially disrupted by the COVID-19 epidemic, the state program has continued in 2020 and 2021 to address the national Pb problem.

### 4.6. Next Steps for Identification of Pb Sources and Value of Specific Interventions

Several initiatives are essential for identifying Pb sources in more detail, including using Pb isotope ratio analysis in blood and environmental samples to confirm specific sources of Pb contributing most to the elevated BLL [34]. A related approach is developing additional intervention programs based on particular Pb source information, which may differ across geographic areas, or providing dietary supplements leading to targeted interventions in specific subgroups or areas. It has been suggested that remediation efforts should focus on harm reduction and contribute to efforts to plan for optimal social development with appropriate consideration of environmental aspects [35,36].

## 5. Conclusions

In the Georgian setting, we documented a significant reduction of BLL in children over a relatively short time period.

This can be attributed to a range of proactive interventions by the public health authorities alongside media interest in the country.

We observed most of the reduction in relation to a relatively simple set of interventions which included individual written and verbal communication (see Table 1) together with home visits in the subgroup with highest exposure. The more robust intervention (such as state program which included medical examination) was targeted to the same children and appeared to maintain this BLL reduction, although this could be documented only for a short period of time.

Monitoring is ongoing and will provide an opportunity to evaluate the BLL in a larger sample of children. This could include addressing very significant differences we reported in this study in the prevalence of BLL of public health concern between the Western and Eastern regions of the country. At this time, the reason for this geographical difference, as well as the Pb sources of greater importance to children in Georgia, is unknown. Lead exposure source research is crucial to provide evidence for designing robust, country-wide effective public health interventions in combination with measures already implemented.

Evidence-based, individually tailored, targeted interventions could be designed and should be implemented. On that basis, once the key Pb sources are identified, it may be feasible to achieve a further sustained reduction of BLL among children already identified, and more importantly achieve a reduction among all children exposed.

## Figures and Tables

**Table 1 ijerph-18-11903-t001:** Brief description of the information and advice/recommendations provided to parents.

Lead-Related Topic	Information and Advice Provided to Parents
1. Effects of lead on child’s health	Lead adversely affects mental development such as cognitive skills and learning abilities. Lead can affect hearing, behavior, growth and intellectual development.
2. Main symptoms	Reduced learning ability; low scores in intellectual assessment tests; restricted growth and development; attention and concentration problems; speech-related problems; low academic scores; coordination problems; behavior problems; hyperactivity.
Advice/recommendations provided	
1. Behavior-related advice	Deny access to surfaces with lead paint or with any flaking paint; regularly wash your child’s hands and wash child’s toys; regularly wet-clean your home; avoid use of folk-medicines; pay attention that your child has no access to cosmetic products; protect children from construction and renovation works; protect yourself during house re-construction works from exposure to lead-containing materials; do not smoke around your children; do not let children go near landfills or waste sites; pay attention that your children have no contact with bare soil (do not play with/in soil). If it is possible, provide your children with special playing sand boxes, or arrange to surface bare soil with grass or wood.
2. Diet-related advice	
A diet rich in calcium strengthens bones and increases elimination of lead from the body.	Milk and milk products, such as yogurt and cheese; green leafy vegetables, such as cabbage, turnip, mustard and greens; calcium-fortified food, such as orange juice, soya milk and tofu; canned salmon and sardines.
Iron hinders lead absorption; try to eat food rich in iron.	Lean red meat; iron-fortified cereals, bread and pasta; dried fruits, such asraisins and black plums; kidney beans and lentils.
Vitamin C improves iron absorption in the body, which can replace the lead in the body.	Citrus fruit, such as orange and grapefruit; other fruit, such as kiwi, strawberry, melon; tomato; potato; pepper.

**Table 2 ijerph-18-11903-t002:** Summary of the median changes in Pb concentration (µg/dL) between MICS Survey (2018) and state program (2019).

Region	N	MICS Survey, 2018 Median and IQR Pb Concentration	N	State Program August, 2019 Median and IQR Pb Concentration	N	State Program Most Recent Result (September−December, 2019) Median and IQR Pb Concentration
Overall	423	9.6 [6.8–14.1]	364	7.1 [4.6–11.1]	315	6.8 [4.3–10.6]
Adjara	96	11.0 [7.3–19.8]	84	9.4 [6.5–15.1]	79	10.2 [6.8–16.4]
Guria	99	11.0 [8.6–15.8]	91	8.7 [6.6–12.0]	83	8.8 [5.7–11.8]
Imereti	56	9.3 [6.7–13.5]	52	5.5 [4.2–9.1]	44	6.8 [4.5–10.1]
Kakheti	15	6.5 [5.3–9.1]	11	4.0 [3.0–7.0]	5	2.9 [2.0–3.2]
Mtskheta	18	7.7 [5.9–11.0]	11 *	5.1 [4.6–6.9]	16 *	4.1 [2.9–7.6]
Kvemo Kartli	9	6.8 [5.6–8.2]	8	4.6 [4.1–5.5]	3	4.9 [3.1–9.8]
Samegrelo	57	9.4 [7.4–12.5]	53	6.2 [4.4–8.5]	41	6.6 [4.3–8.0]
Samtskhe−Javakheti	12	8.3 [6.1–15.4]	6 *	9.7 [4.5–16.0]	10 *	7.5 [4.1–10.9]
Shida kartli	38	7.9 [5.9–10.0]	29	4.2 [3.1–5.0]	23	4.7 [3.2–6.8]
Tbilisi	23	8.3 [6.3–10.8]	19	5.2 [3.9–9.6]	11	6.9 [5.3–9.2]

* Number of children providing samples after August increased because some families could not attend August appointments.

**Table 3 ijerph-18-11903-t003:** Parameter estimates and associated statistics.

Effect	N	Estimate	Standard Error	*t* Value	Pr > |t|
Intercept	423 *	2.4	0.08	29.05	<0.0001
East	423	−0.22	0.11	−1.85	0.0641
East * August	423	−0.52	0.05	−11.30	<0.0001
East * Recent	423	−0.56	0.05	−12.13	<0.0001
West * August	423	−0.29	0.03	−11.62	<0.0001
West * Recent	423	−0.33	0.03	−13.03	<0.0001
Intercept	256 **	2.4	0.09	24.870	<0.0001
East	256	−0.14	0.16	−0.920	0.356
East * August	256	−0.39	0.07	−5.690	<0.0001
East * Recent	256	−0.48	0.07	−6.940	<0.0001
West * August	256	−0.26	0.03	−9.080	<0.0001
West * Recent	256	−0.305	0.029	−10.680	<0.0001

* The whole cohort; ** only those tested in August as well as after August.

**Table 4 ijerph-18-11903-t004:** Pairwise comparisons between different BLL analysis points (August 2019 vs. MICS-2018; Recent vs. MICS-2018; August 2019 vs. recent) in West and East parts of Georgia.

Label	N	Estimate	Standard Error	*t* Value	Pr > |t|
East vs. West in August	423 *	−0.23	0.05	−4.3	<0.0001
East vs. West in Recent	423	−0.23	0.05	−4.4	<0.0001
August vs. Recent in East	423	0.04	0.05	0.84	0.4029
August vs. Recent in West	423	0.04	0.03	1.42	0.1558
East vs. West in August	256 **	−0.14	0.08	−1.81	0.0707
East vs. West in Recent	256	−0.18	0.08	−2.36	0.0182
August vs. Recent in East	256	0.09	0.07	1.26	0.2092
August vs. Recent in West	256	0.05	0.03	1.59	0.1099

* The whole cohort; ** only those tested in August as well as after August.

## Data Availability

Data underlying the findings described in the manuscript will not be shared.

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
