# Peer review of "Reduction in Blood Lead Concentration in Children across the Republic of Georgia following Interventions to Address Widespread Exceedance of Reference Value in 2019"

_ijerph, 2021, doi:10.3390/ijerph182211903_

Round 1
Reviewer 1 Report
Congratulations on the job is welcome. I suggest the authors to detail in a small table the provision of information and recommendations. If I do, I would like to review it.
Author Response
Dear Reviewer,
We are very thankful for giving us your feedback on our manuscript and believe that the changes that have been implemented improve the overall reading of our manuscript.
As you suggested we added the table (Table 1: Brief description of the information and advice/recommendations provided to parents) to the section “2. Materials and methods.”
Sincerely Yours
Ekaterine Ruade
Reviewer 2 Report
The authors found a median reduction of BLL between MICS 2018 and first-stage intervention in August 2019 by 2.5 μg/dL and between MICS 2018 and the most recent BLL results by 2.8 μg/dL. The median BLL reduction between August 2019 and the most recent BLL results (0.3 μg/dL) were not statistically significant. These results were important for public health and it is urgent to continue to intervene to reduce Pb exposure.
Author Response
Dear Reviewer,
Thank you very much for revising the manuscript and thanks for your feedback. We believe that the intervention we reviewed is important to improve public health. This intervention continued in 2020-2021.
Truly Yours
Ekaterine Ruadze
Reviewer 3 Report
Introduction
Currently the paper is too weak in terms of framing the paper with respects to other research. Specifically, I see nothing on the health effects of lead exposure.
The article needs to mention and cite some of the major systems it affects like-
1) cardiovascular system
Lanphear, Bruce P., Stephen Rauch, Peggy Auinger, Ryan W. Allen, and Richard W. Hornung. "Low-level lead exposure and mortality in US adults: a population-based cohort study." The Lancet Public Health 3, no. 4 (2018): e177-e184.
Obeng-Gyasi, E., Ferguson, A.C., Stamatakis, K.A. and Province, M.A., 2021. Combined Effect of Lead Exposure and Allostatic Load on Cardiovascular Disease Mortality—A Preliminary Study. International journal of environmental research and public health, 18(13), p.6879.2)
2) Renal system
Harari, Florencia, Gerd Sallsten, Anders Christensson, Marinka Petkovic, Bo Hedblad, Niklas Forsgard, Olle Melander et al. "Blood Lead Levels and Decreased Kidney Function in a Population-Based Cohort." American Journal of Kidney Diseases (2018).
Lin, Ja-Liang, Dan-Tzu Lin-Tan, Kuang-Hung Hsu, and Chun-Chen Yu. "Environmental lead exposure and progression of chronic renal diseases in patients without diabetes." New England Journal of Medicine 348, no. 4 (2003): 277-286.
3) Hepatic system
Can, S., C. Bağci, M. Ozaslan, A. I. Bozkurt, B. Cengiz, E. A. Cakmak, R. Kocabaş, E. Karadağ, and M. Tarakçioğlu. "Occupational lead exposure effect on liver functions and biochemical parameters." Acta Physiologica Hungarica 95, no. 4 (2008): 395-403.
Among others… We need to understand that it is critical to explore this due to leads effects over the life course.
Methods:
Statistics
More justification needed of sample sizes (i.e. power, effects size etc) especially for subsample analysis.
Results:
Results:
Table 1- why did you report median rather than Mean levels, 95 percent confidence intervals and p-values?
Fugure1 is highly distorted and needs to be redone.
Discussion.
This is perhaps the most important section of a paper. As written the discussion does not give enough context to the results from other works and from the findings. More is needed, especially in the context of lead exposure risk for which there is extensive literature.
Author Response
Dear Reviewer,
We are very thankful for giving us your feedback on our manuscript and believe that the changes that have been implemented improve the overall reading of our manuscript.
Below you can find our responses to your comments/questions.
Truly Yours
Ekaterine Ruadze
- English language and style are fine/minor spell check required
Author Reply:
Thank you for revising the manuscript. And thanks for your feedback. The manuscript was further edited by a native English speaker.
- Introduction
Currently, the paper is too weak in terms of framing the paper with respect to other research. Specifically, I see nothing on the health effects of lead exposure.
The article needs to mention and cite some of the major systems it affects like-
1) cardiovascular system
Lanphear, Bruce P., Stephen Rauch, Peggy Auinger, Ryan W. Allen, and Richard W. Hornung. "Low-level lead exposure and mortality in US adults: a population-based cohort study." The Lancet Public Health 3, no. 4 (2018): e177-e184.
Obeng-Gyasi, E., Ferguson, A.C., Stamatakis, K.A. and Province, M.A., 2021. Combined Effect of Lead Exposure and Allostatic Load on Cardiovascular Disease Mortality—A Preliminary Study. International journal of environmental research and public health, 18(13), p.6879.2)
2) Renal system
Harari, Florencia, Gerd Sallsten, Anders Christensson, Marinka Petkovic, Bo Hedblad, Niklas Forsgard, Olle Melander et al. "Blood Lead Levels and Decreased Kidney Function in a Population-Based Cohort." American Journal of Kidney Diseases (2018).
Lin, Ja-Liang, Dan-Tzu Lin-Tan, Kuang-Hung Hsu, and Chun-Chen Yu. "Environmental lead exposure and progression of chronic renal diseases in patients without diabetes." New England Journal of Medicine 348, no. 4 (2003): 277-286.
3) Hepatic system
Can, S., C. Bağci, M. Ozaslan, A. I. Bozkurt, B. Cengiz, E. A. Cakmak, R. Kocabaş, E. Karadağ, and M. Tarakçioğlu. "Occupational lead exposure effect on liver functions and biochemical parameters." Acta Physiologica Hungarica 95, no. 4 (2008): 395-403.
Among others… We need to understand that it is critical to explore this due to leads effects over the life course.
Author Reply:
We added the new paragraph to the introduction section.
The first paragraph of the introduction section in the current version of the manuscript reads as follows:
“Lead (Pb) is a widespread and harmful environmental toxicant [1-3] that causes adverse health effects in children, particularly neurological and neurobehavioral deficits, lower IQ, slowed growth, and anemia [4-7]. In addition, health effects of Pb over the life course have been documented, including adverse effects on the cardiovascular [8,9], renal and hepatic system [10-12].”
We have kept this rather short as the focus of our manuscript is on children and therefore not on chronic effects of lead exposure which we consider to be outside of the scope of this manuscript.
3. Methods:
Statistics
More justification is needed of sample sizes (i.e. power, effects size etc), especially for subsample analysis.
Author Reply:
We added clarifying details to the first paragraph of the “Statistical methods” section.
All children (i) less than 18 years old (ii) tested in the frame of the 2019 state program and (iii) also in the MICS 2018 were included in the statistical analysis. In the frame of the State Program were also children (<18 years old) not tested in the MICS and they were excluded from the analysis, as well as adult pregnant family members.
We did not take a sample of children but rather we included ALL children who were eligible based on the inclusion criteria ((i) Age <18 years, (ii) tested in the frame of the State Program and at the same time (iii) were tested in the MICS-2018. This was part of a very necessary public health intervention and although a power calculation could have been conducted based upon the previous studies reported in Canada and the USA we strongly argue that this would not be appropriate for a number of reasons. The Canadian study was focussed on a population exposed to lead from a smelter and changes in individual cohorts from year to year were highly variable (+0.5 to -4ug/dL, P-value 0.71 to 0.0001) and the North Carolina and Vermont study was based upon an action level of 10ug/dL and the NHANES data described in the final paper describes decreases in the lead over the periods 1976-1980 and 1988 -1991 making the intervention period a minimum length of 8 years. The published studies therefore we feel are not appropriate for making a power calculation for our study. Instead, we adopted the use of a Nationally representative sample to determine whether the state program could elicit a meaningful reduction in BLL. With respect to the continued decline in BLL over the 5 months of the state program, we have added ‘consequently underpowered to detect a statistical difference between August 2019 and last samples’ to the limitations section.
- Results:
Table 1- why did you report median rather than Mean levels, 95 percent confidence intervals, and p-values?
Author Reply:
We consider that reporting median with IQR is the appropriate statistical choice for describing the distribution of blood lead level as biomarker data are not normally distributed.
- Figure1 is highly distorted and needs to be redone.
Author Reply:
We removed figure 1.
- Discussion.
This is perhaps the most important section of a paper. As written the discussion does not give enough context to the results from other works and from the findings. More is needed, especially in the context of lead exposure risk for which there is extensive literature.
Author Reply:
We added to the Discussion some brief text to interpret our results in the context of other similar studies and added some references to the discussion section to make it stronger.
Reviewer 4 Report
Manuscript review
Overall: The project was a worthy public health endeavor. However, the methods are not robust enough to warrant consideration as a scientific research paper for this journal. It would be more appropriate as a public health surveillance report.
Study design: the authors call this a repeated cross-sectional survey, but the results are reported as if it’s a intervention pre/post design. To control for outside influences on lead levels such as increased press coverage of lead in spices, a control group should have been used.
Methods: the two stages of intervention are dissimilar enough that they should not be included in the same analysis. Also, the fact that the elevated group was told to see a physician for a neurologic assessment might have influenced the parents to pay more attention to lead levels. This indicates that participants received different levels of intervention, all but nullifying the ability to analyze the data as a pre and post analysis. Also, children with Pb >5 ug/dL received supplements that might decrease absorption of lead. So which intervention was tested in this analysis? All of them together?
Finally, informed consent was waived. It’s concerning that blood tests were collected without the written consent of the parents of the children.
Author Response
Dear Reviewer,
We are very thankful for giving us your feedback on our manuscript and believe that the changes that have been implemented improve the overall reading of our manuscript.
Below you can find our responses to your comments/questions.
Truly Yours
Ekaterine Ruadze
- English language and style are fine/minor spell check required
Author Reply:
The manuscript was edited, in terms of English language and style by a native English speaker.
- Manuscript review
Overall: The project was a worthy public health endeavour. However, the methods are not robust enough to warrant consideration as a scientific research paper for this journal. It would be more appropriate as a public health surveillance report.
Author reply:
We appreciate that the issue we studied is a public health problem that needed to be addressed, however in addition to surveillance we present a quantitative estimate on the result of several interventions provided by a national network of agencies, and we discuss the limitations, therefore we consider that this is a valid addition to the literature.
- Study design: the authors call this a repeated cross-sectional survey, but the results are reported as if it’s an intervention pre/post design. To control for outside influences on lead levels such as increased press coverage of lead in spices, a control group should have been used.
Author Reply:
We agree that the term “repeated cross-sectional survey” is not accurate. We changed it to a “longitudinal study”, which appropriately describes our analysis. It is longitudinal as we analyzed data collected a few times on the same children (for some children it was collected three times and for others more than three times).
To remove the possible effect of the press coverage (which might be the alternative explanation of our findings) the possible control group would be children with high BLL but not covered by any of the interventions discussed in the paper (strictly speaking the interventions the effect of which we investigated are exposure factors). To select such a control group was neither possible nor ethical considering the high prevalence of elevated BLL in the Georgian children and the well-established adverse consequences of Pb exposure in children above the action level.
In addition, our analysis was done retrospectively. If we would have conducted a prospective study we couldn’t leave children with high BLL without any intervention such as advice on how to reduce the BLL. Secondly, we could not select controls at the moment of analysis i.e. children with high BLL without intervention, which is not ethical.
Also, the main analytical comparison of interest was not between the group of children included and another group, but of the value of BLL in this group at different time points, leading to a longitudinal study. This description is general and could also be termed “repeated measures study” or “panel study” but we prefer to adopt the more general description of the design.
As we have described in the methodology section, the MICS-2018 data has been employed as our baseline data for comparison and though it is appreciated that the decrease in BLL may be a consequence of factors other than the communication strategy employed. Consequently, in the limitations section we have added:
‘As it was not considered ethical to monitor children with elevated BLL and not provide advice on reducing Pb exposure, we also cannot discount entirely that the decline in BLL is not a consequence of other factors.’
- Methods: the two stages of intervention are dissimilar enough that they should not be included in the same analysis. Also, the fact that the elevated group was told to see a physician for a neurologic assessment might have influenced the parents to pay more attention to lead levels. This indicates that participants received different levels of intervention, all but nullifying the ability to analyze the data as a pre and post analysis. Also, children with Pb >5 ug/dL received supplements that might decrease absorption of lead. So which intervention was tested in this analysis? All of them together?
Author Reply:
We had a chance to retrospectively follow children tested in MICS-2018.
After the MCIS-2018 there were two dissimilar public health interventions. We acknowledge the dissimilarity and during the analysis, we did not mix them. First, we looked at how BLL changed after the first intervention which happened before the initiation of the second intervention. There is a clear cut-off between these interventions and we know the exact time and BLL of children at the moment of initiation of the second intervention. In the manuscript, we report and discuss results in three ways: 1. First stage intervention BLL results (August) vs baseline MICS-2018 BLL result; 2. The end of both interventions (recent) BLL result vs. baseline MICS-2018 BLL result, and 3. Second intervention (state program) BLL results vs first intervention (August) BLL results. Such an approach enables us to answer the following questions: 1. How much BLL was reduced as a result of the first intervention. 2. How much BLL was reduced as a result of both (this is particularly interesting for the national, Georgia audience, and decision-makers) interventions and 3. What kind of changes happened as a result of the second stage intervention?
We were not in a position nor was our intention to separately look at a specific aspect of a public health intervention (i.e. supplement, advice, etc), but rather we took a holistic approach. The question we wanted to investigate was: Does this public health intervention as it is designed (we give a very detailed description of both interventions) work to reduce BLL? We suspect that it is unlikely that a single intervention would work for all children, therefore, given the high proportion of Georgian children with BLL above the action level this multicomponent approach aimed at reducing Pb exposure from different sources as well as addressing the potential consequences of poor diet was agreed to be the most appropriate to achieve some degree of success.
- Finally, informed consent was waived. It’s concerning that blood tests were collected without the written consent of the parents of the children.
Author Reply:
The blood tests were collected following ethical approvals: 1. During the MICS-2018 (institutional review board: 8th August 2018 (NCDC IRB # 2018-033). and 2. Within the frame of the State Program. In the framework of the State Program blood is usually collected in the medical establishments. According to the Law of Georgia on Patient Rights [reference 24, English version of the Law] adequate written informed consent was obtained. In both cases, the informed consent form was obtained from a parent or guardian of the child.
The statement we had in the previous version of the manuscript has led to confusion “Patient consent was waived due to the fact that was conducted the analysis of the secondary data and collection of consents was not feasible. In addition, this research has public health implications as the findings can lead to the improvement of the population health “in the section “Informed Consent Statement”. This statement only relates to the data analysis part i.e. we did not collect informed consent forms additionally for data analysis. However, collection of data (i.e. blood samples) and enrolment onto the programs did not happen without the written informed consent of a parent or guardian.
The statement about informed consent in the current version reads as follows: “According to the Law of Georgia on Patient Rights written informed consent is required within medical establishments, and was obtained. The current study analysis BLL data were collected in two stages: first in the framework of the MIC-2018 survey and second in the State Program. Written informed consent from children’s parents or guardians were collected for each blood sample, both those included in the MICS-2018 and those covered by the State Program. In the framework of the State Program blood was collected in the medical establishments. Parents/guardians were free to withdraw their consent at any time. Ethics approval for MICS -2018 was obtained on the 8th August 2018 (NCDC IRB # 2018-033). Institutional review board approved the presented study in June 2020 (NCDC IRB #2020-026) [24].”
Round 2
Reviewer 3 Report
The article is much improved.
Reviewer 4 Report
No further comments.